# Higher Prevalence of Nonsense Pathogenic *DMD* Variants in a Single-Center Cohort from Brazil: A Genetic Profile Study That May Guide the Choice of Disease-Modifying Treatments

**DOI:** 10.3390/brainsci13111521

**Published:** 2023-10-28

**Authors:** Vitor Lucas Lopes Braga, Danielle Pessoa Lima, Tamiris Carneiro Mariano, Pedro Lucas Grangeiro de Sá Barreto Lima, Ana Beatriz de Almeida Maia, Wallace William da Silva Meireles, Kécia Tavares de Oliveira Pessoa, Cristiane Mattos de Oliveira, Erlane Marques Ribeiro, Paulo Ribeiro Nóbrega, André Luiz Santos Pessoa

**Affiliations:** 1Division of Pediatry, Hospital Infantil Albert Sabin, Fortaleza 60410-794, CE, Brazil; vitorlucas.vlb@gmail.com (V.L.L.B.); anabeatrizdam@gmail.com (A.B.d.A.M.); 2Division of Geriatry, Walter Cantidio University Hospital, Federal University of Ceara, Fortaleza 60430-372, CE, Brazil; dra.daniellelima@gmail.com; 3Division of Neurogenetics and Neuromuscular Disorders, Hospital Infantil Albert Sabin, Fortaleza 60410-794, CE, Brazil; dratamirismariano@gmail.com; 4Faculty of Medicine, Federal University of Ceara, Fortaleza 60020-181, CE, Brazil; pedro.lucas@alu.ufc.br; 5Secretaria Estadual de Saúde do Distrito Federal, Sobradinho 73010-124, DF, Brazil; wallacewsmeireles@gmail.com; 6Division of Nutrition, Hospital Infantil Albert Sabin, Fortaleza 60410-794, CE, Brazil; kecia.oliveira@yahoo.com.br; 7Division of Physiotherapy, Hospital Infantil Albert Sabin, Fortaleza 60410-794, CE, Brazil; olicrism@gmail.com; 8Division of Genetics, Hospital Infantil Albert Sabin, Fortaleza 60410-794, CE, Brazil; erlaneribeiro@yahoo.com.br; 9Division of Neurology, Walter Cantidio University Hospital, Federal University of Ceara, Fortaleza 60430-372, CE, Brazil; 10Campus Parque Ecológico, Centro Universitário Christus, Fortaleza 60160-230, CE, Brazil; 11Albert Sabin Children’s Hospital, Fortaleza 60410-794, CE, Brazil; andrepessoa10@yahoo.com.br; 12Faculty of Medicine, State University of Ceará (UECE), Fortaleza 60714-903, CE, Brazil

**Keywords:** neuromuscular diseases, dystrophin, Duchenne muscular dystrophy, muscular dystrophies, myopathy, therapeutics, genetic profile, pathogenic variant

## Abstract

Dystrophinopathies are muscle diseases caused by pathogenic variants in *DMD,* the largest gene described in humans, representing a spectrum of diseases ranging from asymptomatic creatine phosphokinase elevation to severe Duchenne muscular dystrophy (*DMD*). Several therapeutic strategies are currently in use or under development, each targeting different pathogenic variants. However, little is known about the genetic profiles of northeast Brazilian patients with dystrophinopathies. We describe the spectrum of pathogenic *DMD* variants in a single center in northeast Brazil. This is an observational, cross-sectional study carried out through molecular-genetic analysis of male patients diagnosed with dystrophinopathies using Multiplex Ligation-dependent Probe Amplification (MLPA) followed by Next-Generation Sequencing (NGS)-based strategies. A total of 94 male patients were evaluated. Deletions (43.6%) and duplications (10.6%) were the most recurring patterns of pathogenic variants. However, small variants were present in 47.1% of patients, most of them nonsense variants (27.6%). This is the largest South American single-center case series of dystrophinopathies to date. We found a higher frequency of treatment-amenable nonsense single-nucleotide variants than most previous studies. These findings may have implications for diagnostic strategies in less-known populations, as a higher frequency of nonsense variants may mean a higher possibility of treating patients with disease-modifying drugs.

## 1. Introduction

The dystrophin gene (*DMD*) is the largest gene described in human beings (locus Xp21.2-p21.1; OMIM#310200), spanning more than 2.5 million bp of the genomic sequence, which corresponds to about 0.1% of the total human genome or about 1.5% of the entire X chromosome [1]. RNA transcribed from the dystrophin gene is expressed predominantly in skeletal and cardiac muscle, with lower expression in brain tissue [1]. Since the discovery of *DMD* in 1986 [2], many different types of pathogenic variants have been described in dystrophinopathies, including large deletions and duplications, single-nucleotide variants and small rearrangements [3]. Data regarding DMD prevalence in Brazil are scarce and not yet estimated [4].

Dystrophinopathies are skeletal muscle diseases caused by pathogenic variants in *DMD* [5]. They represent a spectrum of muscle diseases that include the phenotypes of an asymptomatic increase in serum creatine phosphokinase (CK); muscle cramps with myoglobinuria; *DMD*-associated dilated cardiomyopathy—OMIM # 302045; Duchenne muscular dystrophy (DMD)—OMIM # 310200; and Becker muscular dystrophy (BMD)—OMIM # 300376 [6]. DMD is the most severe form of the disease and the most common neuromuscular disorder in childhood, affecting 1 in 3500 live male births [7], while BMD is a milder disease with a later onset and slower progression compared to DMD [8].

Several therapeutic strategies for the correction of dystrophinopathies are currently being developed, and some of these treatments, including stop-codon-readthrough drugs or exon skipping using antisense oligonucleotides, are undergoing clinical trials or are used by patients under conditional approval around the world [9]. Conducting studies on local cohorts in order to know the populational profile of *DMD* variants could lead to healthcare improvements and facilitate access to emerging pathogenic-variant-specific treatments [3].

Little is known about the genetic profiles of northeast Brazilian patients with dystrophinopathies [5,10]. Thus, this paper aims to describe the spectrum of pathogenic variants of dystrophinopathies in a single center in northeast Brazil.

## 2. Materials and Methods

This is an observational, cross-sectional study carried out through a molecular-genetic analysis of male patients diagnosed with dystrophinopathies, followed on an outpatient basis between September 2021 and July 2023 at a Tertiary Pediatric Hospital in the State of Ceará, Brazil. All data reported in this study were collected after obtaining free and informed consent from legal guardians and, when possible, from patients. This paper was approved by the local Ethics Committee under register number 78568717.0.0000.5042. Male patients with progressive myopathy and elevated CK levels (more than a 3-fold increase in CK) were screened for dystrophinopathies. Patients who had pathogenic *DMD* variants and were regularly followed up were included in this study. Patients diagnosed through muscle biopsy without molecular confirmation were excluded.

Genomic DNA was obtained from buccal swab samples. Different genetic analysis techniques were used. Genotyping was carried out in different laboratories, including both public (academic) and commercial CLIA-certified laboratories. Multiplex Ligation-dependent Probe Amplification (MLPA) using the MRC Holland (Amsterdam, The Netherlands) SALSA MLPA P034-B1-1013 and P035-B1-1013 kits was performed initially in most patients to evaluate duplications and deletions. Multiplex-PCR-amplified products were separated by capillary gel electrophoresis in an ABI3500xl Genetic Analyzer (Applied Biosystems, Foster City, CA, USA ). Next-Generation Sequencing (NGS)-based strategies were performed for patients who did not present exon deletions or duplications. Variants were annotated using the lastest version of ANNOtate VARriation (ANNOVAR) software and Ensembl Variant Effect Predictor (VEP) and then filtered using custom R-scripts. Filtered variants were rare or absent in control population databases (gnomAD exome and gnomAD genome). To detect previously reported variants, we used Pubmed in addition to ClinVar. The in silico predictors used were Polyphen, Genomic Evolutionary Rate Profiling (GERP) score, Combined Annotation Dependent Depletion (CADD) and SpliceAI. Variants were classified according to American College of Medical Genetics (ACMG) guidelines for interpretation of variants [11]. Assessment of disruptions in the reading frame caused by large duplications and deletions was performed using the LOVD exonic deletions/duplications reading-frame checker [12].

All treatments, including corticosteroids, cardiovascular drugs and ataluren, were administered at the discretion of the attending physicians. There was no specific treatment protocol for this study, and information regarding treatment was provided by the attending physicians.

We used IBM SPSS 20 to organize and analyze clinical data from patients. Categorical variables are expressed in absolute numbers and percentages. Continuous data are represented by mean values and standard deviations. Comparative statistical analysis was performed to evaluate genotype–phenotype correlations. Pearson’s chi-square test was used to compare categorical variables.

## 3. Results

In total, 96 patients were assessed in this study, and 94 patients from 87 unrelated families fulfilled the inclusion criteria. The median age at the last follow-up was 12 years (SD 5.1 years). Two patients had muscle biopsies compatible with dystrophinopathy without genetic confirmation and were excluded from the analysis. The phenotypes found were Duchenne muscular dystrophy (DMD) in eighty-six patients and Becker muscular dystrophy (BMD) in eight patients. The relative frequencies of all variants found in this study are shown in Figure 1.

We found thirty different patterns of deletion and ten patterns of duplication, summarized in Figure 2. Thirty-six DMD patients (36/86; 41.86%) and five BMD (5/8; 62.5%) patients presented exon deletions. In DMD patients, the isolated deletion of exon 45 and the deletion of exons 48 to 52 were the most common sites, and both occurred in 4.65% of patients (*n* = 4/86). Other exons with increased frequencies of deletions were exons 45–50 (*n* = 3/86, 3.49%), 48–50 (*n* = 2/86, 2.32%), 46–52 (*n* = 2/86, 2.32%) and 12–44 (*n* = 2/87, 2.32%). Most exon deletions in patients with DMD were clustered around a distal “hot spot” involving exons 45–53 (25/36, 69.44%). In BMD patients, there were no clear patterns of deletions.

There were no recurring patterns of exon duplication. Only one patient with a BMD phenotype had an exon duplication, while the remaining nine exon duplications were found in DMD phenotypes.

Small variants (single-nucleotide variants, frameshift or small indel changes—previously reported as point mutations) were present in 45.7% (*n*= 43/94) of patients. Most of these pathogenic variants were observed in DMD patients, where 47.7% (*n* = 41/86) of patients had small pathogenic variants. Nonsense variants were the most common small variants in DMD patients (*n* = 23/41, 56.1%), followed by frameshift (*n* = 11/41, 26.8%) and splice-site variants (*n* = 7/41, 17.1%) (Table 1). Previously described splice-site variants were the only small variants found in BMD patients. Missense pathogenic variants were not found.

We identified 32 different small pathogenic or likely pathogenic variants (Table 2). The previously described nonsense variants c.8038C > T p.(Arg2680*) and c.453T > G p.(Try151*) were reported in three related patients each, and the splice-site variant c.3603 + 3A > T was reported in four related patients.

Five novel pathogenic and likely pathogenic variants were found in this study.

### Clinical Characteristics

The mean age of the 94 patients included in the analysis was 12 years at the last follow-up (standard deviation of 5.1 years). The clinical phenotype was DMD in 91.5% of patients and BMD in 8.5%. The occurrence of “in-frame” rearrangements had a strong correlation with a milder Becker muscular dystrophy phenotype (Table 3).

Loss of ambulation occurred in 38.2% of patients for whom we had information on motor functions. The mean age of ambulation loss was 9.45 years (standard deviation of 2.21 years). Information on left ventricular ejection fraction on echocardiogram was available for 39 patients. Of these, 30.8% had a left ventricular ejection fraction under 55%. A total of 19.1% of patients were using angiotensin converter enzyme inhibitors (ACEIs) or angiotensin receptor blockers (ARBs) for cardiomyopathy. Clinical information is summarized in Table 4.

Most patients were using steroids continuously (98.9%), all of whom received deflazacort. The only patient who did not use steroids was diagnosed at the age of 3 months through familial screening and was still asymptomatic at the age of 9 months.

Regarding disease-modifying treatments, 22 patients (23.40%) were amenable to treatment with ataluren. Of these, 13 patients (*n* = 13/22, 59.1%) were using ataluren, and 2 other patients received a prescription of ataluren and are currently awaiting access to treatment. The variants present in these patients amenable to treatment with ataluren were all nonsense variants. The most common variants were p.(Arg2680*), p.(Arg2095*) and p.(Arg145*), each one in two unrelated families (Table 5). No other disease-modifying drugs were used by any patients.

## 4. Discussion

Our study enrolled patients with dystrophinopathies from a single center in northeast Brazil. The northeastern region of Brazil comprises nine states and covers an area of 1.554.257 km^2^, housing a population of 54.6 million people, accounting for 27.1% of the total Brazilian population, making it the second most populous region in Brazil. The northeastern population is ethnically composed of 62.5% mixed-race individuals, 29.2% white individuals, and 7.8% black individuals [13]. This region is reported to have high levels of inbreeding [14].

In our sample, the most common pathogenic variants in *DMD* were large deletions (43.6%), most located within the distal “hotspot” between exons 45 and 52. Large deletions were also the most recurring type of pathogenic variants (58.2%) in a previous Brazilian study, and hotspots encompassed exons 46 to 52 [5]. It has been suggested that deletion/duplication hotspots within the *DMD* gene are likely to be similar among populations, with a major hotspot around exons 45–52 and a minor hotspot around exons 3–19, because copy-number variations in these regions are more likely to result in an out-of-frame RNA encoding for critical dystrophin domains [15]. However, some authors have proposed that the distribution and frequency of deletions in the *DMD* gene can vary due to population-specific intronic sequences [16,17] (Alu sequences, short tandem repeats, matrix-associated regions, replication origins, microhomology regions) [15] that may lead to a predisposition to preferential deletion breakpoints. Large deletions were also reported as the most common variants in *DMD* in other South and Central American countries [18,19,20].

Duplications are a common pattern in *DMD* variants and were present in 10.6% of patients in our cohort. In most cases, duplications result in the impairment of gene function due to the extra exon copy disrupting the reading frame of *DMD*. Similarly to deletions, duplication breakpoints vary among families [21]. Exon 7 duplication was the most recurring pattern in the present cohort. In a previously published study of patients with *DMD* duplications, a predominance of duplications was observed in exons 3–7 [15].

We found no missense variants in the present cohort. Missense variants were seldom reported in previous DMD studies. In a Brazilian cohort of 144 patients, only 1 had this type of pathogenic variant [5]. Similarly, in the TREAT-NMD DMD global database, missense variants correspond to less than 1% of all pathogenic *DMD* variants [3]. Thus, our data are compatible with previous studies in this regard.

According to the TREAT-NMD DMD database, approximately 10–15% of DMD patients have a nonsense single-nucleotide variant in *DMD* [3]. A large population-based Canadian study reported only 67 patients with nonsense variants out of a total of 773 (a percentage of 8.6%), almost all of them with a Duchenne muscular dystrophy phenotype [22]. The present study detected a prevalence of nonsense variants of 24.5%, which is slightly higher. The percentage of nonsense variants in our population was also significantly higher when compared to other South American countries, where the presence of nonsense single-nucleotide variants varied from 11.6 to 16.7% [5,18,23]. A single-center study conducted in Mexico, however, found similar results regarding the prevalence of nonsense pathogenic variants (*n* = 11/49; 22.45%) [24]. When compared to data from the Brazilian population, the incidence of nonsense variants in the population of Ceará is similar to that described in Salvador, which was reported to be 27.7%, but different from that found in a Brazilian multicentric cohort, which was 12.4% [5]. Ceará and Salvador, the capital of Bahia, are geographically located in the northeast region of Brazil and share a similar colonization process, which could be the reason for the similar findings.

In the present series, there were only three families with more than one patient identified: four related patients had the splice-site variant c.3603 + 3A > T, three patients presented the nonsense variant c.8038C > T; p.(Arg2680*) and three patients from the same family presented the nonsense variant c.453 T > G; p.(Try151*). We have not systematically screened asymptomatic family members due to difficulties in access to genetic testing. Only one patient had a pre-symptomatic diagnosis.

The splice-site variant c.3603 + 3A > T was found in four patients from the same family. This variant was related to classic DMD in three patients (one of them with cor triatriatum) and one asymptomatic patient diagnosed by neonatal screening due to a positive family history. This variant is not present in population databases (gnomAD). This sequence change is located in intron 26 of the *DMD* gene and does not change the encoded amino acid sequence of the DMD protein. RNA analysis indicates that this variant induces altered splicing and may result in an absent or disrupted protein product. Variants that disrupt the consensus splice site are a relatively common cause of aberrant splicing, and functional studies have shown that this variant results in the activation of a cryptic splice site and introduces a premature termination codon [25] leading to the nonsense-mediated decay of the resulting mRNA. However, residual wild-type splicing was also detected, which could explain the milder Becker rather than Duchenne phenotype in previously reported patients. This variant has been previously reported as pathogenic (ClinVar Variation ID: 409882), being observed in individuals with Becker muscular dystrophy (BMD) [22,25,26]. A very large Canadian study with 773 patients found only 2 patients with this variant, both with a Becker muscular dystrophy phenotype [22]. Remarkably, all symptomatic patients with this variant in the present series had a classical Duchenne phenotype. We have found no previous reports of DMD phenotypes with this variant, and the previous report of residual wild-type splicing would lead us to suppose that this variant should lead to a Becker muscular dystrophy phenotype in most cases. Unfortunately, we could not perform muscle biopsies with immunostaining for dystrophin in these patients to assess whether there was residual dystrophin staining. Moreover, the fact that all of these patients were from a single family precludes us from making hypotheses regarding a possibly higher frequency of this variant in our population.

Considering only unrelated patients, three nonsense variants (c.8038 C > T; c.433 C > T; and c.6283 C > T) were the most common, each appearing in two independent families. These variants result in the premature interruption of *DMD* translation, resulting in absent or disrupted proteins. These variants have been previously reported as pathogenic (ClinVar IDs: 217213, 11225, 94697), causing a DMD phenotype [27]. The phenotypes reported are consistent with our findings, as all our patients presented with DMD.

Regarding the differential diagnosis, as NGS is scarcely available in our region, we performed NGS only in patients with clinical findings typical of DMD, which decreased the number of differential diagnoses achieved by NGS. We had only two patients who underwent NGS and found variants in other disease-related genes, one case with *TK2*-related mitochondrial DNA depletion syndrome, myopathic form (TK2-MDS), and one with *COL4A2*-associated myopathy. Other studies with more inclusive criteria for NGS in patients with myopathy found multiple LGMD loci as differential diagnoses [24]. Despite the possibility of a somewhat similar phenotype, LGMD usually presents with lower CK levels compared to DMD, but NGS may help differentiate these conditions in selected cases.

Genetic testing in patients with dystrophinopathies allows for an accurate diagnosis and assumes importance when considering the emergence of treatment measures based on *DMD* variants. By characterizing *DMD* variants in local cohorts, it is possible to benefit populations by directing appropriate clinical management and facilitating access to emerging pathogenic-variant-specific treatments [9,28]. Accurate molecular diagnosis, given by the identification and precise characterization of deleterious variants, is crucial for dystrophinopathy patients to confirm the clinical presumptive diagnosis, to access the specific standard of care and to determine eligibility for the available pharmacogenetic treatments [18,29]. Thus, the importance of this study lies in describing the genetic variability of DMD patients in a previously unreported population.

Over the last few years, there has been considerable development of diagnostics and therapeutics for DMD, and several therapeutic strategies for the correction of a reading-frame shift in DMD patients are currently being developed or are already available for use [9]. Readthrough therapies utilize small molecules to interact with ribosomes, which leads to the insertion of an alternative amino acid at the point of the premature termination codon to allow translational readthrough so that a relatively functional dystrophin protein can be generated [30]. Exon-skipping therapy is based on the removal of an additional *DMD* gene exon neighboring a patient’s deletion to convert an “out-of-frame” pathogenic *DMD* variant to an “in-frame” variant [31].

In our population, at least 32.6% of patients are candidates for some type of available therapy. Compared to the TREAT-NMD database, in our series, 27.6%% of patients are candidates for the most readily available readthrough therapy for nonsense variants versus only 10% in the database. As for exon skipping for pathogenic variants and duplications, eight patients in the present cohort could receive exon-skipping therapy: three patients for exon 45, two for exon 51 and one patient for exon 53. Despite these findings, only 15.9% of patients were receiving disease-modifying therapies at the last study follow-up, a fact that highlights the difficult journey from clinical symptoms to diagnosis and access to treatment in underdeveloped countries, even in specialized centers for neurogenetic diseases. We believe that raising awareness of this potentially treatable disease in the medical literature may help to improve public policies aimed at providing a timely diagnosis and treatment to these patients.

The clinical characteristics of our cohort were somewhat similar to those in previous studies on dystrophinopathies, with a mean age of 12 years at the last follow-up and a predominance of the clinical phenotype of Duchenne muscular dystrophy over Becker muscular dystrophy (91.5% × 8.5%). As this study was performed in a reference center for child neurology, most patients with dystrophinopathies had an early-onset Duchenne muscular dystrophy phenotype.

Ambulation loss is an expected feature of the natural history of dystrophinopathies and tends to occur more prematurely in DMD patients compared to Becker [32]. The overall mean age for ambulation loss in our series was 9.45 years old. This value is slightly lower than in most previous reports showing an age of onset of wheelchair dependency varying from 10 to 13 years in treatment-naïve patients [33,34]. The reasons for this somewhat early compromise of ambulatory functions are difficult to speculate at this point, but a lack of access to rehabilitation therapies, such as physical therapy, adequate treatment for scoliosis and other deformities and possibly a delay in diagnosis and the start of steroid treatment might all be involved in the early loss of ambulation seen in this population.

Only 39 patients in the present series had access to an echocardiogram. Of these, 30.8% had a reduction in left ventricle ejection fraction (LVEF < 55%). Given the age range of our patients, it was expected that a larger portion of our cohort would show some level of cardiac involvement. The literature reports that 59% of patients with dystrophinopathies show cardiac dysfunction by the age of 10, and almost all of them present with cardiomyopathy in adulthood [35]. There is evidence pointing to a trend toward an increase in the prevalence of cardiac dysfunction in DMD and BMD patients since support therapies permit a decrease in early deaths related to respiratory failure, and late complications of the disease become more common due to increased survival [36]. Left ventricle ejection fraction (LVEF) is reported to be reduced in an age-dependent proportion of DMD patients, usually starting to decrease to less than 55% by the age of 11, reaching a plateau between 15 and 25 years and presenting a sharp descent after the age of 25 years [37]. This pattern of decrease could not be assessed in our study due to the difficulty in performing serial echocardiogram evaluations.

Most patients in the present series received treatment with steroids. All of these patients were treated with deflazacort. Oral steroid treatment for Duchenne muscular dystrophy is recommended to all patients, regardless of the pathogenic variant profile, due to its nonspecific effects. The corticosteroids deflazacort and prednisone/prednisolone are standards of care for the treatment of DMD [38]. Both drugs appear to improve muscle strength and slow disease progression [39,40,41,42], and their increased use in early patient management is credited with changing the natural history of the disease [29,41]. Deflazacort was the first drug approved by the United States Food and Drug Administration (FDA) in 2017 for the treatment of DMD at a 0.9 mg/kg/d once-daily oral dose [43,44]. Treatment with deflazacort was significantly associated with improvements in muscle strength scores after 12 weeks of therapy [43]. In Europe, this drug has already been in use for some time [45]. The Brazilian consensus and recommendations for DMD treatment suggest the use of deflazacort in the face of the potential for fewer side effects with prolonged therapeutic use [46]. However, some centers have been using prednisone or methylprednisolone as an off-label alternative [41]. Although there are no studies providing a direct comparison between deflazacort and prednisone, one recent research study in the Cooperative International Neuromuscular Research Group (CINRG) natural history cohort found that deflazacort-treated individuals had a higher median age of ambulation loss than prednisone/prednisolone-treated patients [41], and a meta-analysis of the placebo arms (standard of care) of recent randomized controlled trials showed that deflazacort-treated patients vs. prednisone/prednisolone-treated patients experienced, on average, lower declines in 6 min walking distance, rise from supine, 4-stair climb, and the North Star Ambulatory Assessment linearized score [47]. In our center, all patients received deflazacort in accordance with FDA guidelines.

Regarding disease-modifying therapies, 13 patients (13.14%) were referred to treatment with ataluren. A recent interim analysis (2022) of the STRIDE Registry of real-world patients treated with ataluren compared with the CINRG Duchenne Natural History Study (2015–2022), including 307 patients from 14 countries, showed a significant delay in age at loss of ambulation of 4 years (*p* < 0.0001), as well as a significant delay in age at decline to a predicted forced vital capacity < 60% of 1.8 years (*p* = 0.0021) and to a predicted forced vital capacity < 50% of 2.3 years (*p* = 0.0207) [48]. A recent meta-analysis including data from two randomized clinical trials (ClinicalTrials.gov: NCT00592553; NCT01826487) has shown a significant effect of ataluren on slowing disease progression versus placebo in patients with prolonged use of the medication (>48 weeks), more remarkably in patients who presented at baseline with a 6 min walk distance of 300m or more to less than 400m [49].

Despite these results favoring ataluren use, an FDA advisory panel has rejected the application of ataluren for DMD treatment. This decision was made based on a randomized clinical trial showing no significant improvement in the prespecified primary outcome of an improvement in 6MWT in patients with previously worsening ambulation despite an improvement in 6MWT in the entire cohort analysis [50,51].

The European Medicines Agency (EMA) has historically approved the commercialization and use of ataluren for DMD patients. However, EMA is reviewing this decision and, at the time of writing of this article, the agency has not recommended renewing the marketing authorization of ataluren, although this drug is currently registered for use in Europe. The new position was mostly based on a lack of significant efficacy observed in Phase 3 Clinical Trial Study 041 (NCT03179631) for the prespecified subgroup of patients who had a progressive decline in their ability to walk. In this subgroup of patients, the study did not show a statistically significant difference between ataluren and placebo in terms of the distance patients could walk in six minutes after 18 months of treatment; the EMA suggested that failure to achieve significance in that group might mean that the difference observed in the broader population involved in the study may be due to chance [52,53]. In addition, an analysis of patient registry data comparing the health outcomes of patients who had been treated with ataluren for an average of 5.5 years with those of patients who had not received ataluren showed a delay in the loss of walking ability; however, the EMA committee could not draw conclusions from these data due to methodological issues and uncertainty linked to the indirect comparison: the registry patients were compared with a historical cohort. Waiting for definite conclusions regarding the therapeutic efficacy of ataluren in delaying muscle disease progression in DMD patients with amenable mutations, in Europe the drug currently continues to be prescribed.

This study has some limitations. First, we used a convenience sample, which can lead to some bias. Second, the samples were processed in more than one laboratory. Also, two patients with muscle biopsies compatible with dystrophinopathy and no identified pathogenic variants in WES were excluded from the analysis, but whole-genome sequencing was not performed, and it is possible that these patients had DMD due to variants not identifiable with WES. However, we believe this is the largest single-center cohort in Latin America and the only one to include patients from the northeastern state of Ceará, where previous studies have documented significant inbreeding [29] and many novel ultra-rare disease-causing pathogenic variants have been described [30,31,32]. The implementation of strategies aimed at improving access to genetic investigations in isolated populations from underdeveloped countries may lead to the discovery of rare variants [54], expand the disease spectrum in previously described diseases [55,56] and help in differentiating disease-modifying variants from genetic variations caused by differences in ancestrality.

Populations with European ancestry are usually over-represented in international multicentric disease databases. Knowing the genetic profiles of dystrophinopathies in large cohorts of populations with unique genetic backgrounds, such as northeast Brazil, could help to identify the more common variants in specific populations and to develop public health strategies directed at providing access to specific treatment for amenable pathogenic variants.

## 5. Conclusions

We have described the spectrum of pathogenic variants in a large single-center DMD cohort in Latin America. The genotype spectrum was somewhat different from what has been described in other populations and in the TREAT-NMD database. Notably, we have found an increased percentage of nonsense variants in our population, which are amenable to treatment with available readthrough therapies. This finding may allow the development of public health policies aimed at increasing diagnostic efforts for DMD and reinforce the need to perform NGS techniques in all patients with negative MLPA studies due to a possibly higher proportion of treatment-amenable cases in our population.

## Figures and Tables

**Figure 1 brainsci-13-01521-f001:**
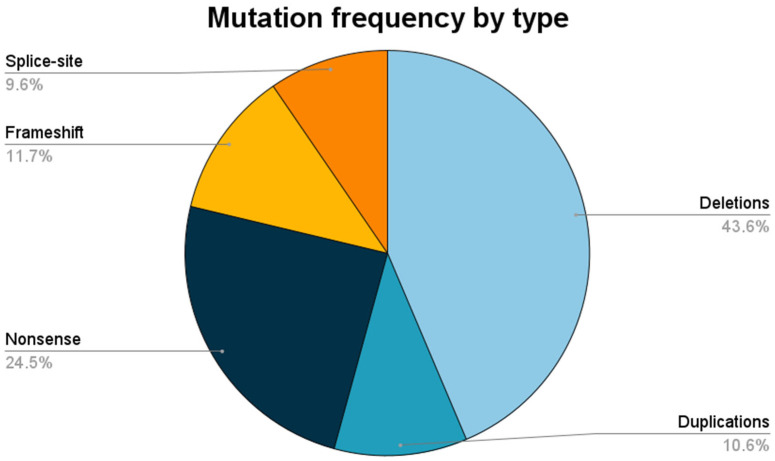
Relative frequencies of pathogenic variants by type.

**Figure 2 brainsci-13-01521-f002:**
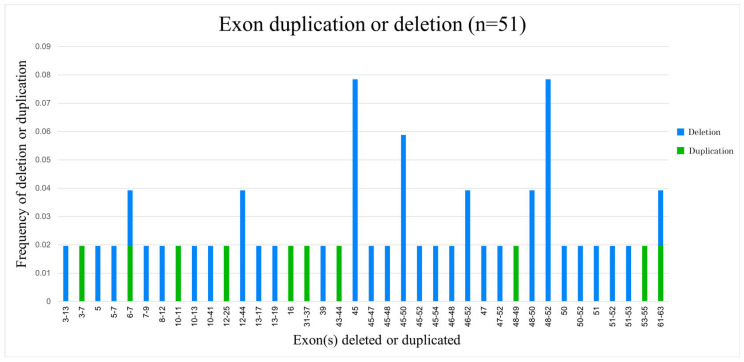
Relative frequencies of rearranged exons in deletions and duplications.

**Table 1 brainsci-13-01521-t001:** Frequencies of different pathogenic *DMD* variants in patients with dystrophinopathies.

Type of Pathogenic Variant	DMD (*n* = 86; 91.5%)	DMB (*n* = 8; 8.5%)	All Patients (*n* = 94; 100%)
LARGE			
Deletions	36 (41.9%)	5 (62.5%)	41 (43.6%)
Duplications	9 (10.5%)	1 (12.5%)	10 (10.6%)
SMALL			
Nonsense	23 (26.7%)	0	23 (24.5%)
Frameshift	11 (12.8%)	0	11 (11.7%)
Splice-site	7 (8.1%)	2 (25.0%)	9 (9.6%)

**Table 2 brainsci-13-01521-t002:** Profiles of small pathogenic variants in unrelated northeastern Brazilian families with dystrophinopathies.

Mutation	Families	Nucleotide Change	Protein Change	ClinVar	dbSNP
Nonsense	2	c.8038 C > T	p.(Arg2680*)	217213	rs863225011
Nonsense	2	c.433 C > T	p.(Arg145*)	11225	rs128626235
Nonsense	2	c.6283 C > T	p.(Arg2095*)	94697	rs398124008
Nonsense	1	c.453 T > G	p.(Try151*)	803948	rs1603437254
Splice-site	1	c.3603 + 3A > T		409882	rs1060502615
Nonsense	1	c.3151C > T	p.(Arg1051*)	94576	rs398123929
Frameshift	1	c.4314_4315delAA	p.(Arg1439Serfs*6)	94619	rs398123950
Nonsense	1	c.6292 C > T	p.(Arg2098*)	11260	rs128626250
Nonsense	1	c.8608C > T	p.(Arg2870*)	94810	rs398124074
Nonsense	1	c.8944G > A	p.(Arg2982*)	11211	rs128625229
Nonsense	1	c.9337C > T	p.(Arg3113*)	94839	rs398124092
Frameshift	1	c.141dupG	p.(Arg48Glufs*41)	565437	rs1569533965
Frameshift	1	c.2552_2553insA G > GT	p.(Asn851Lysfs*17)	◊	◊
Nonsense	1	c.10011C > A	p.(Cys3337*)	*	*
Frameshift	1	c.3295_3296delCA	p.(Gln1099Asp fs*11)	*	*
Frameshift	1	c.5131del	p.(Gln1711Serfs*10)	1685728	
Nonsense	1	c.Gln3037C > T	p.(Gln3037*)	1322249	
Nonsense	1	c.133C > T	p.(Gln45*)	196372	rs794727499
Frameshift	1	c.3533_3536delAAGA	p.(Glu1178Glyfs*22)	803892	rs1603633864
Frameshift	1	c.9269_9270delAG	p.(Glu3090Alafs*)	803806	rs1603253563
Frameshift	1	c.3185_3192delinsTTTGTAT	p.(Lys1062llefs*10)	*	*
Frameshift	1	c.3396delA	p.(Lys1132Asnfs*20)	*	*
Frameshift	1	c.6986del	p.(Lys2329Serfs*9)	455927	rs398124040
Nonsense	1	c.8744 G > A	p.(Trp2915*)	803815	rs1603222922
Nonsense	1	c.9248G > A	p.(Trp3083*)	◊	◊
Nonsense	1	c.5646 C > A	p.(Tyr1882*)	◊	◊
Nonsense	1	c.6276C > A	p.(Tyr2092*)	618598	rs1569555987
Splice-site	1	c.5740-1G > T		1365986	
Splice-site	1	c.9362-1G > C		1685709	
Splice-site	1	c.2804-1del		496616	rs1557374667
Splice-site	1	c.2169-1G > A		803918	rs1603635331
Splice-site	1	c.9286 + 2delT		*	*

◊ Previously reported variants absent in ClinVar. * Novel variants.

**Table 3 brainsci-13-01521-t003:** Correlation between genotypic indel spectrum and phenotype.

Genotypic Spectrum		Phenotype		*p* Value ^1^
		DMD (*n* = 45; 88%)	DMB (*n* = 6; 12%)	
Duplication	In-frame	3 (7%)	1 (17%)	<0.05
	Out-of-frame	6 (13%)	0	
Deletion	In-frame	4 (9%)	5 (83%)	
	Out-of-frame	32 (71%)	0	

^1^ *p* value was assessed using Pearson’s chi-square test.

**Table 4 brainsci-13-01521-t004:** Dystrophinopathy patients’ clinical characteristics.

Characteristic	*n* (*n*%)	Mean (SD)
Age at last follow-up ^1^		12 (5.1)
Loss of ambulation ^2^	21 (38.2)	
Age at loss of ambulation ^1^		9.45 (2.21)
Low LVEF in echocardiogram study ^2^	12 (30.8)	
ACEI or ARB	18 (19.1)	

^1^ Ages are expressed in years. ^2^ Percentages are reported based on the total of patients who had clinical information available regarding this outcome. LVEF: left ventricle ejection fraction; ACEI: angiotensin converter enzyme inhibitor; ARB: angiotensin receptor blocker.

**Table 5 brainsci-13-01521-t005:** Single-nucleotide variants in unrelated families with dystrophinopathies amenable to treatment with ataluren.

Variant	*n* (*n*%)
p.(Arg2095*)	2 (10.5)
p.(Arg2680*)	2 (10.5)
p.(Arg145*)	2 (10.5)
p.(Try151*)	1 (5.26)
p.(Arg1051*)	1 (5.26)
p.(Arg3113*)	1 (5.26)
p.(Trp2915*)	1 (5.26)
p.(Trp3083*)	1 (5.26)
p.(Arg145*)	1 (5.26)
p.(Arg2098*)	1 (5.26)
p.(Arg2982*)	1 (5.26)
p.(Gln3037*)	1 (5.26)
p.(Gln45*)	1 (5.26)
p.(Arg2870*)	1 (5.26)
p.(Cys3337*)	1 (5.26)
p.(Tyr1882*)	1 (5.26)

All variant descriptions were based on the DMD_NM_004006.2 reference sequence.

## Data Availability

All data are included in the manuscript or Appendix A.

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
