# Peer review of "Higher Prevalence of Nonsense Pathogenic *DMD* Variants in a Single-Center Cohort from Brazil: A Genetic Profile Study That May Guide the Choice of Disease-Modifying Treatments"

_brainsci, 2023, doi:10.3390/brainsci13111521_

Round 1

Reviewer 1 Report

Comments and Suggestions for Authors

The article is devoted to the study of a cohort of Brazilian patients with Duchenne/Becker muscular dystrophy. The methods used in the work are adequate for such a study. Such studies are interesting not only from a practical point of view - for planning the volumes of those in need of one or another form of etiotropic or pathogenetic therapy, but also in the scientific plan of studying the differences in the molecular etiology of MDD/MDB and the causes of these differences.

However, I have a few questions and comments to the authors.

Major remarks:

1. The design of the study is not very clearly presented. Lines 102-103 mention certain inclusion criteria, but what exactly they were is not written.

2. The features of cohort formation are not specified. From the text of the work, it seemed that only patients with a confirmed DNA diagnosis were included in it. I wonder if there were other patients and what causes of the disease these patients had. What did they find at WES?

3. It is necessary to indicate whether there were "blood - relatives" in the group of patients. There is a small mention of this in the discussion section for option C.3603+3A>T and that's it. Information about whether repeated mutations have been detected in blood relatives or in unrelated patients is extremely important.

4. It is unclear why the authors evaluate the genotype/phenotype correlation by the TYPE of mutation, and not by the effect. It would be wiser to separate deletions and duplications by effect: with and without frame shift.

Minor remarks:

1. Please clarify what geographically or by other signs means the region "northeast Brazilian". The authors need to describe whether it has population features (I found this information only in the discussion in lines 336-337.

2. In my opinion, Table 4 is the latest and does not structure or supplement the information from the text in any way. I propose to move the table on small variants of the DMD gene from the sup. to the main text.

3. With such a cohort size, it is impractical to talk about frequent mutations. I suggest changing the wording to "recurring mutations".

In general, I liked the work, it is holistic..

Author Response

Date: 10/10/2023

Manuscript No. brainsci-2625002

Title: "Higher prevalence of nonsense DMD pathogenic variants in a single-center cohort from Brazil: A genetic profile study with possible treatment repercussions.”

Dear Editors,

We are very pleased to review and resubmit our manuscript and comment on the suggestions and critics made by the reviewers. We sincerely hope to clarify pending questions raised by the reviewers.

Reviewer comments are raised by yellow, and changes performed are highlighted in red in the main manuscript. 

Reviewer 1:

The article is devoted to the study of a cohort of Brazilian patients with Duchenne/Becker muscular dystrophy. The methods used in the work are adequate for such a study. Such studies are interesting not only from a practical point of view - for planning the volumes of those in need of one or another form of etiotropic or pathogenetic therapy, but also in the scientific plan of studying the differences in the molecular etiology of MDD/MDB and the causes of these differences.

However, I have a few questions and comments to the authors.

Major remarks:

  1. The design of the study is not very clearly presented. Lines 102-103 mention certain inclusion criteria, but what exactly they were is not written.

Response: Thank you for that suggestion. We have provided clear inclusion criteria now, as follows: “Male patients with progressive myopathy and high CK levels were screened for dystrophinopathies. Only patients with pathogenic DMD variants, alive at the time of genetic analysis and who were regularly followed up were included in this study. Patients diagnosed through muscle biopsy but without molecular confirmation were excluded.”

  1. The features of cohort formation are not specified. From the text of the work, it seemed that only patients with a confirmed DNA diagnosis were included in it. I wonder if there were other patients and what causes of the disease these patients had. What did they find at WES?

Response: The reviewer makes an important point. We have now clarified the cohort formation under methods as follows: “Male patients with progressive myopathy and high CK levels were screened for dystrophinopathies. Only patients with pathogenic DMD variants, alive at the time of genetic analysis and who were regularly followed up were included in this study. Patients diagnosed through muscle biopsy without molecular confirmation were excluded.” Regarding differential diagnosis through WES, as NGS is scarcely available in our region, we performed NGS only in patients with clinical findings typical of DMD, which decreased the number of differential diagnoses achieved by NGS. We had only two patients who underwent NGS and found variants in other disease-related genes, one case with TK2-associated myopathy and one with COL4A2 associated myopathy. This is now clear in the text. 

  1. It is necessary to indicate whether there were "blood - relatives" in the group of patients. There is a small mention of this in the discussion section for option C.3603+3A>T and that's it. Information about whether repeated mutations have been detected in blood relatives or in unrelated patients is extremely important.

Response: In the present series there were only three families with more than one patient identified: 4 related patients had the splice-site variant c.3603+3A>T, 3 presented the nonsense variant DMD:c.8038C>T;p.Arg2680* and 3 patients from the same family presented the nonsense variation DMD:c.453 T>G;p.Try151*. We have not systematically screened asymptomatic family members due to difficulties in access to genetic testing. Only one patient had a pre-symptomatic diagnosis. This is now clear in the text.

  1. It is unclear why the authors evaluate the genotype/phenotype correlation by the TYPE of mutation, and not by the effect. It would be wiser to separate deletions and duplications by effect: with and without frame shift.

Response: We entirely agree with that suggestion. We have now divided duplications and deletions in “in-frame” and “out-of-frame” and used this classification to test for genotype-phenotype correlations.

Minor remarks:

  1. Please clarify what geographically or by other signs means the region "northeast Brazilian". The authors need to describe whether it has population features (I found this information only in the discussion in lines 336-337.

Response: We entirely agree with that suggestion. We have now specified that “Our study enrolled patients with confirmed DMD or BMD in a single center in Northeastern Brazil. The Northeast region of Brazil is comprised of nine states and covers an area of 1,554,257 square kilometers. It is home to approximately 54.6 million people, accounting for 27.1% of the total population, making it the second most populous region in Brazilian territory. The Northeastern population is ethnically composed of 62.5% mixed-race individuals, 29.2% white, and 7.8% Black. This region is reported to have high levels of inbreeding.”

  1. In my opinion, Table 4 is the latest and does not structure or supplement the information from the text in any way. I propose to move the table on small variants of the DMD gene from the sup. to the main text.

Response: We entirely agree. We have sent this table to the main text. Thank you for that suggestion.

  1. With such a cohort size, it is impractical to talk about frequent mutations. I suggest changing the wording to "recurring mutations".

Response: Thank you for that comment. We have corrected that.

Sincerely,

Paulo Ribeiro Nóbrega, MD, MSc, PhD

Reviewer 2 Report

Comments and Suggestions for Authors

In this manuscript, Lopes Braga et al., describe an apparent high frequency of nonsense variants in DMD gene as dystrophinopathy-causing genotypes in a sample of Brazilian male patients. Description of identified genotypes is in general well-organized, however several flaws identified along the text, preclude its publication in the original form.

Major concerns:

-        Please recalculate the documented allelic frequencies, as in the discussion section is mentioned that several patients included are relatives (lanes 214-215 and 236-237). This is of utmost importance to include only those UNRELATED patients to achieve realistic allelic frequencies (i.e. the allelic frequency for c.3603+A>T clearly is a not real figure).

-        In the material and methods section, the authors must include a careful description of the treatments received for their patients (age at the start of treatment, motor evaluations before and during treatments, etc.), as these are further discussed (corticosteroid, cardiovascular drugs, AONS, and ataluren). These aspects although important, are further barely discussed, then seem to contribute to a lesser extent to the key message of this manuscript.

-        Interestingly, despite that it is referred that 98.9% of the studied patients using corticosteroid therapy, the mean SD for loss of ambulation (9,45 years old) does not differ from the expected natural history for dystrophinopathies.

-        In fact, you mentioned that all your patients received deflazacort by FDA guidelines. Can you discuss deeply this important aspect?

-        The author must disclose any conflict of interest with the pharmaceutical companies producing Ataluren and AONs, as well as include the rationale and the current FDA posture of prescribing these drugs. This aspect if discussed, must be included since the introduction and material&methods sections, as suggested.

Minor criticisms:

-        Abstract: According to the material and methods section, not all patients negative for DMD gene deletions or duplications were subjected to WES, as some of these patients seem to be subjected to neuromuscular gene panels, stated the material and methods. Please clarify (i.e. generalize as NGS-based strategies?).

-        Lanes 56-57: Please include the MIM ID´s for DMD-associated dilated cardiomyopathy, Duchenne muscular dystrophy (DMD) and Becker muscular dystrophy (BMD) phenotypes.

-        Please specify the type of biological sample type employed to obtain the genomic DNA of studied patients.

-        According to HGVS guidelines, the term "mutation" are discouraged, please consider to employ along text the following term "pathogenic variant" or "pathogenic change".

-        Table 2: Please consider the term "Genotypic DMD spectrum" over "Mutation pattern".

-        Lanes 301 to 320 could be deleted, as this paragraph does not discuss your results.

-        Lane 123 and 132: The term "Point mutations" seems inadequate (it means only a single nucleotide change, and does not include the small frameshift or indel changes), so please change it to "small changes or variations".

-        Table 4: Please change the term "point mutations" to "nonsense single nucleotide changes". Please write the change at the protein level according to HGVS nomenclature guidelines [i.e. p.Arg2680* must be written as p.(Arg2680*)]. Also, would be desirable to include the respective change at the cDNA level and the employed reference sequence.

-        For the most common herein identified single nucleotide variants, and also for those described in Table 4, it would be desirable to include their available dbSNP and CliVar ID´s. I.e. c.3603+3A>T: dbSNP:rs1060502615, ClinVar (https://preview.ncbi.nlm.nih.gov/clinvar/variation):409882).

-        Strikingly, the application of MLPA and NGS-based technologies only achieved the identification of diagnostic DMD genotypes, and none with other neuromuscular or muscular dystrophy phenotypes such as limb-girdle muscular dystrophies. This could be attributed to a very careful clinical selection of their patients, although other series applying the similar NGS-based strategies have been identified as other underlying muscular dystrophies, some of them are sometimes clinically difficult to distinguish from dystrophinopathies, especially if those were associated with a previous normal DMD gene MLPA result (see PMID: 31671740). Can you discuss more this aspect?

-        Please unify the figure of nonsense variants 24.4% lane 205 vs. 25.5% in Figure 1. This reported proportion seems to be higher than that described in other Central and South American countries, although it is very similar to that reported in a Mexican population sample (22.5% of all DMD-causing genotypes, see PMID: 31671740).

-        The supplementary material is not available.

Comments on the Quality of English Language

OK!

Author Response

Date: 10/10/2023

Manuscript No. brainsci-2625002

Title: "Higher prevalence of nonsense DMD pathogenic variants in a single-center cohort from Brazil: A genetic profile study with possible treatment repercussions.”

Dear Editors,

We are very pleased to review and resubmit our manuscript and comment on the suggestions and critics made by the reviewers. We sincerely hope to clarify pending questions raised by the reviewers.

Reviewer comments are raised by yellow, and changes performed are highlighted in red in the main manuscript. 

Reviewer 2:

In this manuscript, Lopes Braga et al., describe an apparent high frequency of nonsense variants in DMD gene as dystrophinopathy-causing genotypes in a sample of Brazilian male patients. Description of identified genotypes is in general well-organized, however several flaws identified along the text, preclude its publication in the original form.

Major concerns:

-        Please recalculate the documented allelic frequencies, as in the discussion section is mentioned that several patients included are relatives (lanes 214-215 and 236-237). This is of utmost importance to include only those UNRELATED patients to achieve realistic allelic frequencies (i.e. the allelic frequency for c.3603+A>T clearly is a not real figure).

Response: We entirely agree. Thank you very much for this suggestion. We have changed that in the table and in the discussion. As there was no family recurrence among duplications and deletions, the allelic counts for these variants did not change.

-        In the material and methods section, the authors must include a careful description of the treatments received for their patients (age at the start of treatment, motor evaluations before and during treatments, etc.), as these are further discussed (corticosteroid, cardiovascular drugs, AONS, and ataluren). These aspects although important, are further barely discussed, then seem to contribute to a lesser extent to the key message of this manuscript.

Response: We entirely agree. Thank you very much for this suggestion. We have added the following paragraph to “Methods”: All treatments, including corticosteroids, cardiovascular drugs, and ataluren were administered at the discretion of the attending physicians. There was no specific treatment protocol for this study and information regarding treatment was provided by the attending physicians.

-        Interestingly, despite that it is referred that 98.9% of the studied patients using corticosteroid therapy, the mean SD for loss of ambulation (9,45 years old) does not differ from the expected natural history for dystrophinopathies.

Response: That is correct. The early loss of ambulation was probably related to delayed diagnosis and poor access to rehabilitation and physical therapy, as is now stated in the discussion.

-        In fact, you mentioned that all your patients received deflazacort by FDA guidelines. Can you discuss deeply this important aspect?

Response: We entirely agree. Thank you very much for this suggestion. We have added a discussion regarding deflazacort and its approval by FDA.

-        The author must disclose any conflict of interest with the pharmaceutical companies producing Ataluren and AONs, as well as include the rationale and the current FDA posture of prescribing these drugs. This aspect if discussed, must be included since the introduction and material&methods sections, as suggested.

Response: We entirely agree. Thank you very much for this suggestion. We have added the conflicts of interest of Dr. André Luiz Santos Pessoa as speaker for PTC Therapeutics, Sarepta Therapeutics, and Roche. We also added a discussion regarding ataluren approval/rejection by FDA and EMA.

Minor criticisms:

-        Abstract: According to the material and methods section, not all patients negative for DMD gene deletions or duplications were subjected to WES, as some of these patients seem to be subjected to neuromuscular gene panels, stated the material and methods. Please clarify (i.e. generalize as NGS-based strategies?).

Response: We have changed that to NGS-based strategies. Thank you very much for this suggestion.

-        Lanes 56-57: Please include the MIM ID´s for DMD-associated dilated cardiomyopathy, Duchenne muscular dystrophy (DMD) and Becker muscular dystrophy (BMD) phenotypes.

Response: We have provided MIM numbers for these conditions. Thank you very much for this suggestion.

-        Please specify the type of biological sample type employed to obtain the genomic DNA of studied patients.

Response: Genomic DNA was obtained from buccal swab samples. This is now clear in the text. Thank you.

-        According to HGVS guidelines, the term "mutation" are discouraged, please consider to employ along text the following term "pathogenic variant" or "pathogenic change".

Response: We entirely agree. Thank you very much for this suggestion. We have corrected that as suggested. As many previous papers have discussed these pathogenic variants with the previous name of “point mutations” we have made this clear in the text.

-        Table 2: Please consider the term "Genotypic DMD spectrum" over "Mutation pattern".

Response: We entirely agree. Thank you very much for this suggestion.

-        Lanes 301 to 320 could be deleted, as this paragraph does not discuss your results.

Response: We would like to discuss with the reviewer the possibility of maintaining this paragraph, as we discuss the use of steroids by our patients and the other reviewers have suggested improving on that discussion. We have added a discussion regarding the FDA approval of deflazacort, which was the only steroid used in our patients, and the widespread use of prednisone or prednisolone in these patients.

-        Lane 123 and 132: The term "Point mutations" seems inadequate (it means only a single nucleotide change, and does not include the small frameshift or indel changes), so please change it to "small changes or variations".

Response: We entirely agree. Thank you very much for this suggestion.

-        Table 4: Please change the term "point mutations" to "nonsense single nucleotide changes". Please write the change at the protein level according to HGVS nomenclature guidelines [i.e. p.Arg2680* must be written as p.(Arg2680*)]. Also, would be desirable to include the respective change at the cDNA level and the employed reference sequence.

Response: We entirely agree. Thank you very much for this suggestion. We have provided the reference sequence of all changes in the subtitles and changed variant notations according to that suggestion.

-        For the most common herein identified single nucleotide variants, and also for those described in Table 4, it would be desirable to include their available dbSNP and CliVar ID´s. I.e. c.3603+3A>T: dbSNP:rs1060502615, ClinVar (https://preview.ncbi.nlm.nih.gov/clinvar/variation):409882).

Response: We entirely agree. We have included the ClinVar number of cited variants. Thank you very much for this suggestion.

-        Strikingly, the application of MLPA and NGS-based technologies only achieved the identification of diagnostic DMD genotypes, and none with other neuromuscular or muscular dystrophy phenotypes such as limb-girdle muscular dystrophies. This could be attributed to a very careful clinical selection of their patients, although other series applying the similar NGS-based strategies have been identified as other underlying muscular dystrophies, some of them are sometimes clinically difficult to distinguish from dystrophinopathies, especially if those were associated with a previous normal DMD gene MLPA result (see PMID: 31671740). Can you discuss more this aspect?

Response: Thank you very much for this suggestion. As NGS is scarcely available in our region, we performed NGS only in patients with clinical findings typical of DMD, which decreased the number of differential diagnoses achieved by NGS. We had only two patients who underwent NGS and found variants in other disease-related genes, one case with TK2-associated myopathy and one with COL4A2 associated myopathy. This is now clear in the text. 

-        Please unify the figure of nonsense variants 24.4% lane 205 vs. 25.5% in Figure 1. This reported proportion seems to be higher than that described in other Central and South American countries, although it is very similar to that reported in a Mexican population sample (22.5% of all DMD-causing genotypes, see PMID: 31671740).

Response: We have included this interesting Mexican paper in the discussion regarding NGS for differential diagnosis.

-        The supplementary material is not available.

Response: Thank you very much for this suggestion. The only table that was in supplementary material has now been moved to the main manuscript per the reviewer´s suggestion.

Sincerely,

Paulo Ribeiro Nóbrega, MD, MSc, PhD

Reviewer 3 Report

Comments and Suggestions for Authors

The study presented in this article focuses on the detection of mutations in the dystrophin gene as part of the diagnosis for Duchenne Muscular Dystrophy (DMD) in a unique Brazilian population. The results of this study hold great significance for researchers with similar interests. The originality of the study is remarkable, and the conclusions drawn in the manuscript are largely accurate. 

MINOR

A suggestion is to include information regarding the prevalence of DMD specifically in Brazil within the introduction.

Additionally, it is questionable to exclude patients who had muscle biopsies consistent with dystrophinopathy but did not undergo genetic confirmation. It should be discussed that not all diagnostic approaches, such as whole-genome sequencing, were employed to detect mutations.

Furthermore, it is advisable to provide information regarding the familial history of repetitive point mutations. This would allow for the investigation of possible connections between patients.

Some mistyping can be found e.g. lines 235, 295 etc.

Author Response

Date: 10/10/2023

Manuscript No. brainsci-2625002

Title: "Higher prevalence of nonsense DMD pathogenic variants in a single-center cohort from Brazil: A genetic profile study with possible treatment repercussions.”

Dear Editors,

We are very pleased to review and resubmit our manuscript and comment on the suggestions and critics made by the reviewers. We sincerely hope to clarify pending questions raised by the reviewers.

Reviewer comments are raised by yellow, and changes performed are highlighted in red in the main manuscript. 

Reviewer 3:

The study presented in this article focuses on the detection of mutations in the dystrophin gene as part of the diagnosis for Duchenne Muscular Dystrophy (DMD) in a unique Brazilian population. The results of this study hold great significance for researchers with similar interests. The originality of the study is remarkable, and the conclusions drawn in the manuscript are largely accurate. 

MINOR

Response: We agree that it would be interesting to include the prevalence of DMD in Brazil. However, data about this condition is scarce and has not been estimated yet. We have made this clear in the text.

Additionally, it is questionable to exclude patients who had muscle biopsies consistent with dystrophinopathy but did not undergo genetic confirmation. It should be discussed that not all diagnostic approaches, such as whole-genome sequencing, were employed to detect mutations.

Response: We agree. We have added the following sentence to the discussion “two patients with muscle biopsy compatible with dystrophinopathy and no identified pathogenic variants in WES were excluded from the analysis. However, whole-genome sequencing was not performed, and it is possible that these patients had DMD due to variants not identifiable with WES.” Thank you very much for pointing that out.

Furthermore, it is advisable to provide information regarding the familial history of repetitive point mutations. This would allow for the investigation of possible connections between patients.

Response: We agree. In the present series there were only three families with more than one patient identified: 4 related patients had the splice-site variant c.3603+3A>T, 3 presented the nonsense variant DMD:c.8038C>T;p.Arg2680* and 3 patients from the same family presented the nonsense variation DMD:c.453 T>G;p.Try151*. We have now mentioned the total number of family and discussed all recurring variants. This is now clear in the text.

Some mistyping can be found e.g. lines 235, 295 etc.

Response: We agree. We have corrected that.

Sincerely,

Paulo Ribeiro Nóbrega, MD, MSc, PhD

Round 2

Reviewer 2 Report

Comments and Suggestions for Authors
The authors addressed satisfactorily most of the criticisms.
However remain some minor details, please ensure the correctness of protein variant nomenclature p.(Arg2680*), instead of p.Arg2680* among other examples (see lanes 145 to 146, lanes 242-243, and Table 2).
Please ensure to include the dbSNP and ClinVar of variants described in Table 2, as previously suggested.
Table 2: what means "CCDS5S395.1" in the c.6974delA variant?

Author Response

Date: 19/10/2023

Manuscript No. brainsci-2625002

Dear Editors, 

We hope that all pending questions and suggestions have been addressed now.

Reviewer 2

The authors addressed satisfactorily most of the criticisms.

However remain some minor details, please ensure the correctness of protein variant nomenclature p.(Arg2680*), instead of p.Arg2680* among other examples (see lanes 145 to 146, lanes 242-243, and Table 2).

Response: We have formatted the protein variant nomenclature as requested.

Please ensure to include the dbSNP and ClinVar of variants described in Table 2, as previously suggested.

Response: We have included the available dbSNP and ClinVar IDs in Table 2.

Table 2: what means "CCDS5S395.1" in the c.6974delA variant?

Response: Typo. It has been corrected.

Sincerely,

Paulo Ribeiro Nóbrega, MD, MSc, PhD